# Beyond Filtering: Adaptive Image-Text Quality Enhancement for MLLM Pretraining

## Abstract

Multimodal large language models (MLLMs) have made significant strides by integrating visual and textual modalities. A critical factor in training MLLMs is the quality of image-text pairs within multimodal pretraining datasets. However, *de facto* filter-based data quality enhancement paradigms often discard a substantial portion of high-quality image data due to inadequate semantic alignment between images and texts, leading to inefficiencies in data utilization and scalability. In this paper, we propose the Adaptive Image-Text Quality Enhancer (AITQE), a model that dynamically assesses and enhances the quality of image-text pairs. AITQE employs a text rewriting mechanism for low-quality pairs and incorporates a negative sample learning strategy to improve evaluative capabilities by integrating deliberately selected low-quality samples during training. Unlike prior approaches that significantly alter text distributions, our method minimally adjusts text to preserve data volume while enhancing quality. Experimental results demonstrate that AITQE surpasses existing methods on various benchmark, effectively leveraging raw data and scaling efficiently with increasing data volumes. We hope our work will inspire future works. The code and the model are available at https://github.com/iclr2025-No9242/AITQE.

## 1 Introduction

Recent advancements in multimodal large language models (MLLMs), which integrate visual and textual modalities, have significantly expanded the capabilities of artificial intelligence (Achiam et al., 2023; Bai et al., 2023; Wang et al., 2024a; Lu et al., 2024; Laurençon et al., 2024b;a; McKinzie et al., 2024; Tong et al., 2024; Lin et al., 2024; Fang et al., 2024; Chen et al., 2024c;b; Zhang et al., 2023; 2024). These models are typically constructed by combining large language models (LLMs) with visual encoders like CLIP (Radford et al., 2021), DFN (Fang et al., 2023), and SigLIP (Zhai et al., 2023). They leverage multimodal pretraining datasets including image-text pairs, interleaved image-text sequences, detailed captions, and optical character recognition (OCR) data.

A crucial component of these datasets is high-quality image-text pairs, where each image is paired with a corresponding textual description. Effective semantic alignment between image and text significantly enhances the model's ability to integrate multiple modalities. Similar to textual data collection, raw image-text pairs are typically sourced from the Internet (e.g., SBU (Ordonez et al., 2011), CC3M (Sharma et al., 2018), CC12M (Changpinyo et al., 2021), LAION (Schuhmann et al., 2021; 2022b)). These raw pairs undergo quality enhancement processes to curate high-quality data, during which a significant proportion of image-text pairs are discarded. For instance, in Qwen-VL (Bai et al., 2023), over 70% of the original image-text pairs are removed. Notably, many high-quality images are excluded solely because their corresponding text lacks semantic alignment (see Figure 1). This filtering approach leads to two drawbacks: (1) the unnecessary loss of a considerable amount of high-quality images, and (2) increased difficulty in scaling pretraining data, as a larger volume of raw data must be collected to achieve the desired quantity of high-quality data. For example, training on 10 million data pairs may require collecting at least 30 million raw data pairs.

Recent studies, like MLM-filter (Wang et al., 2024b), have attempted to filter image-text data by leveraging multimodal language model scoring rather than relying solely on CLIPScore (Radford et al., 2021). However, despite these efforts, existing approaches continue to suffer from the significant drawback of discarding a substantial amount of high-quality images. This limitation raises a

Figure 1: The conventional method (a) discards low-quality samples in raw data, reducing the amount of pretraining data, while our AITQE (b) enhances low-quality samples, retaining the same volume of data for MLLMs pretraining.

critical question: *How can we improve data quality more effectively, ensuring high standards while maximizing the amount of enhanced data?*

In this work, we attempt to address this challenge. We demonstrate that current filter-based quality enhancement techniques inherently discard high-quality images due to low-quality or semantically misaligned text. While maintaining a high threshold for image-text semantic similarity improves the average quality of pretraining data by retaining only high-quality image-text pairs, it also leads to the loss of valuable images. Conversely, lowering this threshold to include pairs with high-quality images but low-quality text can degrade model performance due to the inclusion of semantically misaligned data. This presents a dilemma in balancing data quality and quantity.

To address this limitation, we propose the **A**daptive **I**mage-**T**ext **Q**uality **E**nhancer (**AITQE**), a model designed to dynamically assess and enhance the quality of input image-text pairs. Specifically, for pairs exhibiting low quality-such as low semantic similarity between modalities or subpar linguistic quality, AITQE performs text rewriting, generating high-quality text based on the input image and the raw low-quality text. Additionally, we introduce a contrastive sample learning strategy that deliberately incorporates low-quality image-text pairs during model training to strengthen the model's evaluative capabilities. Unlike existing approaches such as ShareGPT4V (Chen et al., 2023), which focus on generating highly detailed captions that significantly alter the text distribution, our rewriting method makes minimal adjustments to improve low-quality or semantically unrelated text. This strategy maximizes data volume preservation while enhancing overall data quality. Moreover, our approach facilitates a more efficient exploration of scaling-laws in pretraining data.

Our contributions are as follows:

- We introduce an image-text pair quality enhancement method that adaptively improves data quality. The proposed Adaptive Image-Text Quality Enhancer (AITQE) is developed and trained to automatically filter or rewrite image-text pairs based on their assessed quality, thereby producing high-quality data pairs.

- We propose a contrastive sample learning strategy to strengthen the model's evaluative capabilities. By incorporating deliberately selected low-quality samples during training, this strategy enhances the model's ability to discern and effectively improve low-quality image-text pairs.

- We demonstrate that our quality enhancement method surpasses existing approaches on benchmark datasets using the same raw data. This shows that AITQE maximally leverages the information inherent in the raw data and validates that our method scales effectively with increasing volumes of raw data.

## 2 RELATED WORK

### 2.1 IMAGE-TEXT DATA FILTERING

Image-text data filtering is essential for creating high-quality image-text datasets. Traditional filtering techniques predominantly rely on predefined rules, such as language type, length and content of text, image size and content, and the removal of duplicate entries (Byeon et al., 2022; Yu et al., 2023a; Schuhmann et al., 2022a; Chen et al., 2024a). Some approaches employ single-modality

filtering strategies. For example, Xu et al. (2024) analyze CLIP's data curation process by extracting textual entries to filter the dataset. Similarly, DataComp (Gadre et al., 2023) utilizes ImageNet class names for text filtering and leverages class clustering to select data based on visual content overlap with these classes.

In addition to these methods, CLIP-Score (Radford et al., 2021) has become integral to large-scale data construction. LAION (Schuhmann et al., 2021) incorporated CLIP to calculate image-text similarity scores and filter high-quality image-text pairs. Subsequently, many datasets (Schuhmann et al., 2022b) have adopted this approach. DataComp offers a data pool for evaluating various data filtering methods, where several CLIP-based approaches have shown promising performance (Maini et al., 2024; Wang et al., 2024c). Recently, DFN (Fang et al., 2023) trains a CLIP-like filter using high-quality image-text pairs to enhance filtering quality.

Recent studies, such as MLM-Filter (Wang et al., 2024b), employ instruction-tuned MLLM-based filters. This method utilizes four distinct scoring metrics generated by the model to filter data in DataComp. In contrast, our approach enables simultaneous output of all metrics, providing a comprehensive overall evaluation. Furthermore, AITQE does not emphasize filtering data but focuses on enhancing data quality for MLLM training.

## 2.2 CAPTION-BASED IMAGE-TEXT ENHANCEMENT

Caption-based image-text enhancement has emerged as a crucial technique for improving the quality of image-text datasets. LaCLIP (Fan et al., 2023) utilizes LLMs to generate new captions; however, it does not incorporate visual information from the images themselves. To overcome this limitation, Nguyen et al. (2023) employ captioning models to generate new captions for images, using multimodal models such as BLIP-2 (Li et al., 2023) and CoCa (Yu et al., 2022). These regenerated captions are then used to train CLIP models more effectively.

Furthermore, many MLLMs possess inherent caption generation capabilities and integrate recaptioning in their training processes. For instance, ShareGPT4V (Chen et al., 2023) utilizes GPT-4V to generate highly detailed captions and subsequently trains a model to produce high-quality captions solely based on image input. However, such models uniformly rewrite all captions without assessing their original quality, which may introduce biases from the detailed captions and significantly alter the text distribution. In contrast, our approach automatically evaluates the quality of each image-text pair and adaptively decides whether to rewrite the text. AITQE performs minimal adjustments, enhancing low-quality or semantically misaligned captions only when necessary, rather than indiscriminately rewriting all captions. This strategy preserves the diversity of the original data and mitigates potential biases introduced by extensive rewriting. Additionally, by incorporating both the image and the original caption, our method addresses both visual and textual dimensions, ensuring a more balanced and effective enhancement of the dataset.

Regarding the integration of original and synthetic captions, VeCLIP (Lai et al., 2024) and CapsFusion (Yu et al., 2023b) utilize LLMs to merge raw captions with synthetic ones, producing refined captions. However, these methods require a two-stage process: first generating captions and then integrating them with the original ones. This approach not only involves multiple stages but also risks being influenced by low-quality original captions. In contrast, our method adopts an end-to-end framework that evaluates the quality of original captions and performs adaptive rewriting, making minimal adjustments. This strategy reduces the potential adverse effects of poor-quality original captions and provides a more streamlined and effective solution for enhancing image-text datasets.

## 3 BUILDING ADAPTIVE IMAGE-TEXT QUALITY ENHANCER

### 3.1 DATA COLLECTION FOR AITQE

**Overview** The process of building AITQE is illustrated in Fig.2. We collect supervised fine-tuning data from GPT-4o using a unified prompt designed to generate scores based on specific evaluation criteria. Inspired by the MLM-Filter (Wang et al., 2024b), these criteria include text quality, image-text matching, object detail, semantic understanding, and text or chart description. Each score is accompanied by a detailed explanation, along with an overall score and a comprehensive overall explanation. By utilizing structured-outputs feature of GPT-4o, we ensure that the generated data

consistently contained all the required scores and explanations. The prompts used for data collection are provided in the Appendix.

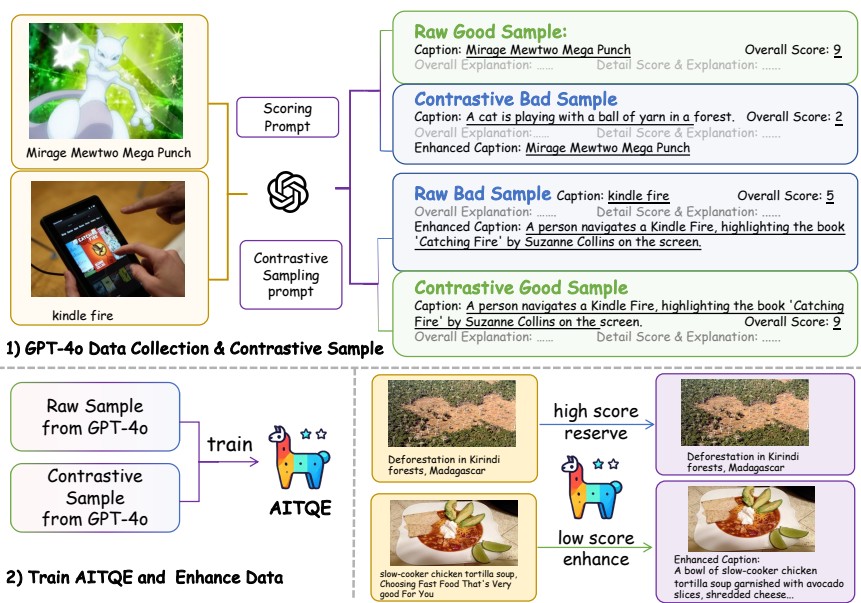

Figure 2: (1) Data collection using GPT-4o in two phases: first scoring raw image-caption pairs with explanation, followed by generation of contrasting quality captions. (2) AITQE model training using collected data and application of AITQE to enhance MLLM pretraining data.

**Data source and First Phase Scoring** We select a single dataset LAION-400M (Schuhmann et al., 2021), as the foundation for all our experiments to maintain consistency and avoid potential discrepancies arising from multiple data sources. We randomly sample 260k instances from this dataset to serve as the basis for GPT annotation, which are exclusively used for fine-tuning our AITQE model. We take inspiration from the criteria outlined in the MLM-Filter (Wang et al., 2024b) but additionally incorporate the identification of potential text or charts within images into our final scoring standards. In designing the prompt, we use a unified prompt that incorporates all these standards, instructing GPT-4o to output an overall integer score ranging from 1 to 10, along with detailed explanations for each criterion. By implementing a JSON schema for structured output, the results could be readily processed into instruction data, enabling our AITQE to produce formatted scoring results. To enhance diversity during processing, we write 20 different sets of instructions and randomly select among them.

**Contrastive Sample Construction** To obtain contrastive samples, we conduct a second round of data collection based on the previously obtained scores. We design two additional prompts:

$P_1$: Instruct GPT-4o to rewrite captions for low-scoring image-text pairs;

$P_2$: Instruct GPT-4o to generate low-quality captions for high-scoring pairs.

Let $I$ be an image, $C$ a caption, and $S(I, C)$ the score of the image-text pair. The newly generated captions $C'$ are limited to a single sentence and scored $S'$:

$$C'_{\text{high}} = P_1(I, C) \text{ where } S(I, C) \text{ is low, and } S'_{\text{high}} = S(I, C'_{\text{high}}),$$
$$C'_{\text{low}} = P_2(I, C) \text{ where } S(I, C) \text{ is high, and } S'_{\text{low}} = S(I, C'_{\text{low}}).$$

After collecting results from both rounds of generation, we construct a dataset $D$ containing negative samples and rewritten captions. Given raw data $D_{\text{raw}} = \{(I_1, C_1), (I_2, C_2)\}$; assuming $S(I_1, C_1)$ is high, generate $C'_{\text{low}}$ with $S'_{\text{low}}$; assuming $S(I_2, C_2)$ is low, generate $C'_{\text{high}}$ with $S'_{\text{high}}$. Then the final constructed dataset $D$ includes the {Image-Text Input: Target Output} pairs as training data:

$$D = \begin{cases} \{(I_1, C_1) : S(I_1, C_1)\}, & \{(I_1, C'_{\text{low}}) : (C_1, S'_{\text{low}})\}, \\ \{(I_2, C_2) : (C'_{\text{high}}, S(I_2, C_2))\}, & \{(I_2, C'_{\text{high}}) : S'_{\text{high}}\} \end{cases}.$$

Finally, the 260k training data is mostly accompanied with contrastive sample, resulting in approximately 520k training data in total. We split them into train and validation set.

## 3.2 DETAILS OF AITQE MODEL

**AITQE model structure**  The AITQE model is composed of SigLIP as the vision encoder, Qwen-2-7B as the language model, and an MLP to connect the visual input with the language model. We employ a two-stage training method. In the first stage, the training data consists of a mix of 558k randomly selected samples from LAION-400M and approximately 690k samples from Cauldron (Laurençon et al., 2024), and we train all model parameters. In the second stage, the training data includes the constructed 520k supervised fine-tuning (SFT) samples mixed with 1.8M samples from Cauldron, and we again train all model parameters.

**Enhance image-text data quality**  We place the rewritten caption (if available) and the overall score at the beginning of target output in the training data, followed by an "$\langle overall \rangle$" indicator after the overall score. This setup facilitates early stopping during the generation phase, reducing the required computing resource and time. If a complete set of scores is needed, the model can output all of them. Thus, when using the model, inputting the image, caption and corresponding instructions will yield the scores, and if the score is low, a rewritten caption will also be provided.

## 3.3 ASSESS THE AITQE MODEL

Table 1: LLaVA trained in two stage, average evaluation results of stage-2 model

| stage-2 model | avg. results |
|---|---|
| run1 | 57.73 |
| run2 | 60.33 |

Table 2: LLaVA trained with our strategy, average evaluation results of stage-1 model

| stage-1 model | avg. results |
|---|---|
| run1 | 61.45 |
| run2 | 60.72 |

To evaluate the AITQE model, we use it to filter or enhance the data for training MLLMs. As the performance of the MLLM reflects the quality of the training data and effectiveness of our AITQE model, it is essential to ensure stable training and evaluation, i.e., if two MLLMs are trained with totally identical setting, the evaluation results should also be close.

**Pre-experiments**  The training strategies of MLLMs are mostly two-stage training following LLaVA (Liu et al., 2024). Thus, we run a preliminary experiments to train LLaVA models using original 558k stage-1 training data and 665k stage-2 data, using identical training settings with LLaVA. Then we evaluate the stage-2 models on five benchmarks: GQA (Hudson & Manning, 2019), OKVQA (Marino et al., 2019), TextVQA (Singh et al., 2019), VQAv2 (Goyal et al., 2017) and SEED-Bench-2 (Li et al., 2024). We run this training-evaluation process twice with totally identical settings, and the results are presented in Tab.1. The average evaluation scores on five benchmarks have a discrepancy of 2.60, which reveals that it can be unstable if we use such training strategy and evaluation process. Therefore, it's vital to adopt a more stable method.

**Training strategy and evaluation setup**  We find out a strategy that stabilizes the training. For the architecture of MLLMs, we also use SigLIP, Qwen-2-7B, and an MLP layer as in AITQE model. During the training stage, we mix image-text pairs with instruction data from Cauldron as a better substitution of the 665k data from LLaVA, and train all parameters of the model, making it possible to evaluate with such stage-1 model. An additional advantage of this strategy is the elimination of a second stage, mitigating potential influence introduced by two-stage training.

For the evaluation, we adopt data from more benchmarks: SEED-Bench-2 (Li et al., 2024), MME (Fu et al., 2023), AMBER (Wang et al., 2023), OKVQA (Marino et al., 2019), VQAv2 (Goyal et al., 2017), DocVQA (Mathew et al., 2021), TextVQA (Singh et al., 2019) and Textcaps (Sidorov et al., 2020). Specifically, for AMBER we test hallucination of attribute, existence, relation and

generative. For MME, we take 4 hallucination related tests (existence, count, position and color) and set the score range to $[0, 100]$. For VQAv2, due to the large amount of testing, we randomly sample 1000 tests. For Textcaps, the CIDEr score is multiplied by 100 for average score counting.

Under this training and evaluation setup, we run another pre-experiments with LLaVA-558k image-text pairs with Cauldron in stage-1, train all parameters in the modal and evaluate with aforementioned evaluation data. The results are in Tab. 2. We can find the difference between two runs are much smaller now, suggesting a more stable training and evaluation.

# 4 EXPERIMENTS

**The foundation of an effective enhancer is a robust filter capable of discerning data quality, thereby enabling targeted improvements to data quality.** Therefore, in Sec. 4.1, we first analyze various filtering approaches using a fixed 2M sample subset from LAION400M. We compare the MLM-Filter's four-criteria scoring with our AITQE model, selecting the top 256K and 558K samples for each method, and using filtered data to train MLLMs.

In Sec. 4.2, we present a comparative analysis of using randomly sampled data versus the same data enhanced by our AITQE method. The datasets of 256K, 558K and 2M samples used in this analysis are identical to those employed in Section 4.1. Additionally, we introduce a larger data pool of 12M samples, which encompasses the 2M dataset used in our previous experiments.

In Sec. 4.3, we compare with ShareCaptioner (Chen et al., 2023) on previously randomly sampled 558K data. The detailed captions by ShareCaptioner are used to replace original captions.

## 4.1 EFFECTIVENESS OF AITQE AS A SCORING FILTER

Table 3: Comparison of MLLMs trained on 256K and 558K samples filtered from a fixed 2M data pool. * denote the process described in Sec. 3.3. MME$^H$ represents the hallucination related test from MME, with scores ranging from 0 to 100. Abbreviations: CTQ (Caption Text Quality), ITM (Image-Text Matching), ODF (Object Detail Fulfillment), SU (Semantic Understanding).

| Benchmarks
*metric* | | **avg.** | SEED2
*ppl* | AMBER*
*ppl/gen* | MME$^H$
*ppl* | OKVQA
*acc* | VQAv2*
*acc* | DocVQA
*acc* | TextVQA
*acc* | Textcaps
*CIDEr* |
|---|---|---|---|---|---|---|---|---|---|---|
| **Model** | **Criteria** | | *256K filtered out of a fixed 2M data pool* | | | | | | | |
| Random | N/A | 58.05(−) | 46.53 | 76.51 | 73.12 | 51.49 | 79.60 | 52.79 | 56.92 | 27.41 |
| MLM-
Filter | CTQ | 60.39 | 48.42 | 80.75 | 78.24 | 52.07 | 78.54 | 55.09 | 58.57 | 31.45 |
| | ITM | 59.35 | 48.26 | 80.72 | 76.98 | 51.81 | 77.42 | 55.79 | 54.41 | 29.39 |
| | ODF | 57.34 | 48.02 | 81.78 | 78.08 | 52.49 | 78.90 | 52.56 | 52.27 | 14.58 |
| | SU | 61.67 | 48.38 | 82.64 | 79.14 | 50.92 | 81.37 | 52.50 | 57.16 | 41.24 |
| | Avg. | 59.69(↑1.64) | 48.27 | **81.47** | **78.11** | 51.82 | **79.06** | 53.99 | 55.60 | 29.17 |
| AITQE | Overall | **65.21**(↑7.16) | **48.73** | 81.15 | 77.77 | **51.84** | 77.86 | **56.99** | **58.29** | **69.07** |
| | | | *558K filtered out of the same fixed 2M data pool* | | | | | | | |
| Random | N/A | 60.07(−) | **47.66** | 79.99 | 72.22 | 52.54 | 78.88 | 55.87 | 55.96 | 37.40 |
| MLM-
Filter | CTQ | 62.67 | 46.76 | 79.94 | 72.84 | 52.31 | 78.20 | 54.11 | 51.21 | 66.01 |
| | ITM | 60.91 | 47.96 | 82.86 | 78.10 | 53.81 | 79.10 | 49.76 | 56.24 | 39.46 |
| | ODF | 61.37 | 47.99 | 81.85 | 78.50 | 52.89 | 80.81 | 52.25 | 58.97 | 37.66 |
| | SU | 62.40 | 46.68 | 82.20 | 78.33 | 52.01 | 80.27 | 57.85 | 59.49 | 42.36 |
| | Avg. | 61.84(↑1.77) | 47.35 | 81.71 | 76.94 | **52.76** | 79.60 | 53.49 | **56.48** | **46.37** |
| AITQE | Overall | 61.89(↑1.82) | 47.57 | **82.66** | **77.12** | 51.99 | **80.73** | **57.00** | 56.01 | 42.04 |
| | | | *558K pretraining data from LLaVA(Liu et al., 2024)* | | | | | | | |
| LLaVA-558K | | 61.09 | 45.80 | 77.73 | 73.77 | 48.99 | 77.39 | 37.93 | 47.98 | 79.12 |

Table 3 presents the comparison of random sampling, MLM-Filter (with four distinct criteria), and our AITQE model. As a reference, we also include the model trained using our strategy with the LLaVA-558k (Liu et al., 2024) image-text dataset (last line in the table).

**AITQE is effective in filtering out high-quality image-text data** In the 256K setting, random sampling exhibits relatively low scores, underscoring the necessity for approaches to enhance the quality training data. The MLM-Filter with four criteria shows varying degrees of improvement, and the average shows an improvement of 1.64. Our AITQE model exhibits the most promising performance, attaining the highest overall average score of 65.21. This represents a substantial improvement of 7.16 points over random sampling and 5.52 points over the average of MLM-Filter, indicating its potential as a robust and versatile filtering method. When compared with the LLaVA-558K trained MLLM, it is worth noting that the AITQE-filtering trained model achieves superior results on several benchmarks and on average despite using less than half the number of training samples. This suggests that our AITQE filtering approach can effectively identify high-quality image-text pairs for MLLM training.

**Lowering the threshold to include more data degrade the model performance** In the 558k setting, we observe that increasing the training data from 256K to 558K random samples resulted in a 2.02 point improvement in the average score of the trained MLLM. This improvement can be primarily attributed to the increased volume of training data. Interestingly, while AITQE-filtering shows improvements over random selection, this gain (+1.82) is less pronounced compared to the 256K scenario (+7.16). And it's important to note that the average score of AITQE under 558k setting is less than that in 256k (−3.32). We posit that this downgrade is due to that the selection process prioritizing higher-scoring samples, the 558K dataset inevitably included the highest-score 256K samples along with additional lower-score data. Consequently, the overall average quality of the 558K dataset decreased. Compared with MLM-Filter, the average performance of AITQE (61.89) is comparable to the average of MLM-Filter (61.84). Compared with LLaVA-558K, AITQE is also slightly better on average (+0.8). This indicates that AITQE provides a consistent and effective filtering approach. These results indicate that increasing data volume through filtering a fixed pool inevitably incorporates lower-quality samples, potentially offsetting the benefits of larger datasets. This highlights the crucial balance between data quantity and quality in MLLM training, especially with limited data resources.

## 4.2 Scaling High-Quality Training Data As an Enhancer

Table 4: AITQE as an image-text quality enhancer for randomly sampled data.

| Data | avg. | SEED2 | AMBER* | MME$^H$ | OKVQA | VQAv2* | DocVQA | TextVQA | Textcaps |
|---|---|---|---|---|---|---|---|---|---|
| 256K rand. | 58.05 | 46.53 | 76.51 | 73.12 | **51.49** | **79.60** | **52.79** | **56.92** | 27.41 |
| 256K-AITQE | **62.34**(↑4.29) | **48.12** | **80.12** | **75.66** | 51.33 | 79.31 | 45.46 | 50.82 | **67.91** |
| 558K rand. | 60.07 | 47.66 | 79.99 | 72.22 | **52.54** | 78.88 | **55.87** | 55.96 | 37.40 |
| 558K-AITQE | **64.12**(↑4.05) | **48.12** | **80.67** | **73.62** | 51.85 | **80.61** | 55.74 | **57.82** | **64.55** |
| 2M rand. | 60.85 | 47.56 | **80.50** | **75.79** | 51.68 | 78.26 | 50.06 | 57.63 | 45.32 |
| 2M-AITQE | **65.35**(↑4.50) | **47.96** | 80.34 | 73.62 | **52.13** | **78.62** | **51.37** | **61.70** | **77.04** |
| 12M rand. | 62.20 | 45.74 | 79.43 | **78.54** | 51.92 | 79.95 | 57.28 | **52.75** | 51.99 |
| 12M-AITQE | **66.90**(↑4.70) | **49.58** | **80.98** | 78.08 | **53.61** | **82.63** | **59.93** | 51.21 | **79.18** |

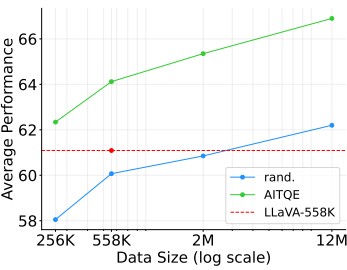

Figure 3: Average performance

**The efficacy of AITQE as a data quality enhancer is evident as we scale up the training data** Beyond its role as a filter for identifying data quality, AITQE's more significant function lies in its capacity to enhance multimodal training data. We present an analysis of the performance gains across different dataset sizes, ranging from 256K to 12M **random** samples. The results are in Tab. 4 and Fig. 3

At the 256K scale, AITQE significantly enhances model performance, with a gain of 4.29 points in the average score. Substantial improvements are observed in tasks such as SEED2, AMBER*, and most remarkably, Textcaps.However, it's worth noting that there are slight decreases in performance for DocVQA and TextVQA at this scale. As we scale to 558K samples, the random sampling shows some improvement over

the 256K baseline, but AITQE-processed data maintains a clear advantage, with an average score improvement of 4.05 , showcasing the consistent benefit of AITQE.

The benefits of AITQE become even more pronounced at the 2M sample scale. Here, we observe that AITQE not only maintains its performance edge but also shows remarkable improvements. The average score for AITQE-enhanced data reaches 65.35, showing a substantial gain of 4.50 points. Notably, Textcaps performance continues to improve, reaching a score of 77.04. At the largest scale of 12M samples, AITQE demonstrates its robustness and scalability. The average score for AITQE-filtered data reaches an impressive 66.90, maintaining a significant gain of 4.70 points. This scale shows improvements across most benchmarks, particularly in SEED2, VQAv2*, DocVQA, and Textcaps.

These results collectively demonstrate the scalability and robustness of our AITQE approach. As the volume of training data increases from 256K to 12M samples, AITQE consistently produces higher-quality datasets that lead to superior MLLM performance. The consistent performance gains across various scales underscore the effectiveness of AITQE in enhancing data quality, ultimately resulting in more capable and versatile multimodal language models.

### 4.3 COMPARE AITQE WITH CAPTIONER

Table 5: Compare with ShareCaptioner on 558K scale

| Model | avg. | SEED2 | AMBER* | MME$^{H}$ | OKVQA | VQAv2* | DocVQA | TextVQA | Textcaps |
|---|---|---|---|---|---|---|---|---|---|
| 558K-ShareCap. | 58.15 | **48.82** | 79.76 | **77.69** | 51.35 | 79.33 | 51.93 | 56.55 | 19.79 |
| 558K-AITQE | **64.12**(↑5.97) | 48.12 | **80.67** | 73.62 | **51.85** | **80.61** | **55.74** | **57.82** | **64.55** |

To further validate the effectiveness of AITQE, we conducted a comparative experiment with Share-Captioner. The results of applying AITQE and ShareCaptioner to a 558K dataset is in Tab. 5.

Our experimental results demonstrate that AITQE significantly outperforms ShareCaptioner in overall performance, with an average score of 64.12 compared to 58.15, representing a substantial improvement of 5.97 points. AITQE shows particular strength in most benchmarks. The most striking difference is observed in the Textcaps, where AITQE outperforms ShareCaptioner with a substantial gap of 44.76 points, highlights AITQE's particular strength in image captioning tasks. However, it's important to note that ShareCaptioner demonstrates stronger performance in MME$^{H}$. This suggests that ShareCaptioner's detailed captions may have specific strengths in handling hallucination tests.

The comparison reveals that while AITQE offers substantial improvements in overall performance and excels in several critical tasks, there remain areas where further refinement could be beneficial.

### 4.4 ANALYSIS OF CONTRASTIVE SAMPLES AND CAPTION REWRITES

Table 6: Performance comparison of MLLMs trained with 256K data, which are filtered by intermediate models that utilize different training strategies.

| Strategy | avg. | SEED2 | AMBER* | MME$^{H}$ | OKVQA | VQAv2* | DocVQA | TextVQA | Textcaps |
|---|---|---|---|---|---|---|---|---|---|
| Base Scorer | 59.88(−) | 47.78 | **81.24** | 78.59 | **51.85** | **81.35** | 53.65 | 57.83 | 26.75 |
| +Contras. Sample | 59.81 | 47.87 | 80.37 | 78.17 | 51.29 | 79.64 | 55.67 | **59.42** | 26.01 |
| +Rewrite Caption | 57.70 | 47.47 | 80.7 | 75.57 | 50.77 | 79.42 | 49.91 | 51.42 | 26.34 |
| +Both → **AITQE** | **65.21**(↑5.33) | 48.73 | 81.15 | 77.77 | **51.85** | 77.86 | **56.99** | 58.29 | **69.07** |

This section presents an analysis of the impact of contrastive sampling and caption rewriting strategies on the performance of our proposed model. The models are used to filter 256K data and Tab.6 presents the resulting scores of each trained MLLM.

**Synergistic Effects of Combined Strategies in building AITQE Model** Our experimental results demonstrate the efficacy of combining contrastive sampling and caption rewriting techniques in enhancing model performance. The base scorer, which is trained with first-round collected GPT scores, achieves an average score of 59.88. When contrastive sampling is applied in isolation, we

observe a marginal decrease in the average score to 59.81. The application of caption rewriting as a standalone strategy results in a more substantial decrease in the average score to 57.70.

Despite the individual shortcomings of these strategies, their combination yields remarkable results. Our proposed AITQE model, which integrates both contrastive sampling and caption rewriting training data, achieves the highest average score of 65.21. This represents a significant improvement of 5.33 points over the base scorer. The AITQE model exhibits consistent performance across most benchmarks, with particularly notable enhancements in Textcaps.

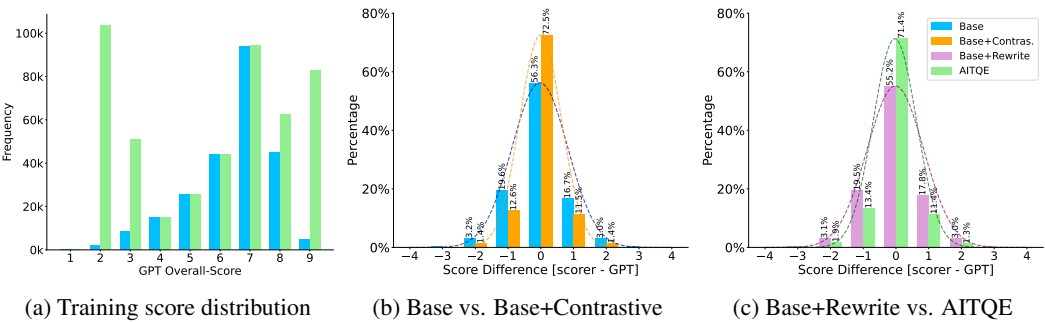

| (a) Training score distribution | (b) Base vs. Base+Contrastive | (c) Base+Rewrite vs. AITQE |

Figure 4: (a) GPT score distribution of base scorer's and AITQE's training set, w/ and w/o contrastive samples. (b) Scoring difference between scorer and GPT score on validation set. (c) The output rewritten percentage of model trained w/ or w/o contrastive samples.

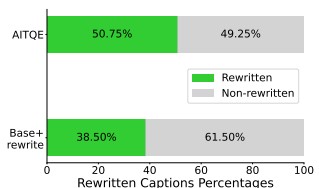

Figure 5: Percentages of rewritten captions in val. set

**Statistical Analysis**   We analyzed the GPT score distribution in the training data and our model's scoring on the validation set, as shown in Fig. 4 and Fig. 5.

In Fig.4a, we analyze the distribution of GPT scores in the training data. The Base scorer training dataset (Base) exhibits a non-uniform distribution but a prominent unimodal peak centered around score 7. This skewed distribution may potentially lead to a model that tends to assign more conservative scores. In contrast, the AITQE training dataset incorporates contrastive samples, resulting in a more diverse distribution. This approach introduces a significant number of low-scoring(2-3) and high-scoring(8-9) samples. The inclusion of contrastive samples serves two primary purposes: **(a) Balancing the data distribution:** The model is exposed to a more diverse set of training examples, which encourages the model to utilize the full spectrum of scores, potentially improving its ability to discriminate between different quality levels. **(b) Reducing potential biases**: Each image is paired with both high and low-quality samples, mitigating potential biases that the model might develop towards specific image characteristics.

In Fig. 4b and Fig. 4c, we analyze the scoring distributions between the models trained with and without contrastive sampled on the validation set, against the original GPT scores. The graph reveals a normal distribution centered around zero for both models. Notably, the Base+contrastive model exhibits a more concentrated distribution, characterized by a smaller standard deviation and a more pronounced peak. This tighter clustering of score differences suggests that the model's predictions align more closely with the GPT scores. This underscores the efficacy of incorporating contrastive samples in the training process, which improves score consistency and accuracy.

Fig. 5 shows the AITQE model, trained with contrastive samples, produces more rewritten captions than the model without. This observation can be attributed to two key factors: increased exposure to caption modifications and as previously discussed, better utilization of the full score range, encouraging lower scores and rewrites when needed. These findings suggest a synergistic relationship between the use of contrastive samples and the generation of rewritten captions. The combination of these two strategies proves to be an effective approach in enhancing the model's capability to critically evaluate and improve image captions. This synergy not only improves the model's discriminative abilities but also its generative capabilities in the context of caption refinement.

## 4.5 QUALITATIVE ANALYSIS

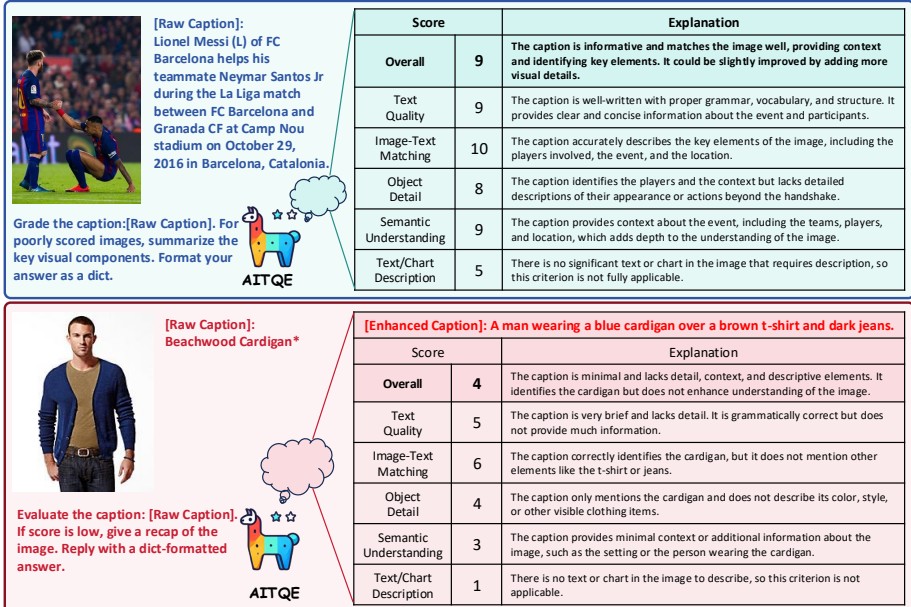

Figure 6: Two examples of AITQE ouputs.

Fig. 6 demonstrates AITQE's effectiveness in evaluating image captions. **Upper:** A high-quality football-related caption (overall score 9), excelling in image-text matching (10) and semantic understanding (9). **Lower:** A low-quality clothing caption (overall score 4), with poor semantic understanding (3) and object detail (4).

AITQE provides numerical scores and detailed explanations for each criterion, enhancing interpretability. For low-scoring captions, it generates an enhanced caption that significantly improves the description's accuracy. This showcases AITQE's ability to discern caption quality across a spectrum of scenarios and its potential to contribute to improved image captioning practices.

## 5 LIMITATIONS

While our study demonstrates the effectiveness of AITQE, two primary limitations warrant consideration. First, our model does not incorporate a Chain of Thought (CoT) approach in its evaluation process (first explanation then score), which could potentially enhance assessment accuracy but at the cost of increased computational resources and time in order to get the final score. This presents a trade-off between efficiency and potential accuracy that merits further investigation. Second, our reliance on a single data source, although beneficial for controlling experimental variables, may limit the generalizability of our findings. Expanding to diverse data sources could provide a more comprehensive evaluation of AITQE's performance and improve its robustness across varied domains. These limitations offer valuable directions for future research, including exploring efficient CoT integration and extending the study to encompass a wider range of multimodal datasets.

## 6 CONCLUSION

In conclusion, this work introduces the Adaptive Image-Text Quality Enhancer (AITQE), an approach that dynamically improves multimodal training data quality while preserving data volume. By overcoming the limitations of traditional filter-based methods, AITQE maximizes the utilization of valuable visual information. Our experimental results demonstrate AITQE's superiority over existing approaches, showcasing its ability to scale effectively with increasing data volumes. This research provides a practical solution to enhance multimodal datasets, potentially leading to more capable MLLM in the future.

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

# A QUALITATIVE ANALYSIS ON TEXTCAPS OUTPUTS

In Sec. 4.3, we compare AITQE with ShareCaptioner, and MLLM trained with AITQE-enhanced data shows better results on 6 out of 8 benchmarks except SEED2 and MME[H]. In Textcaps, the performance gap is huge, and we managed to find out the reason. Here we present a qualitative analysis of the outputs on Textcaps from MLLMs trained with data produced by ShareCaptioner (denoted as $M_S$) and AITQE (denoted as $M_A$).

First, the outputs from $M_S$ show clear patterns like starting with "the image captures...", "the image presents...", "the image showcases...", "in the center of the image...", "in the image...", followed by detailed descriptions of the content. These patterns are introduced by the training data generated by ShareCaptioner and are not present in the reference captions in Textcaps's tests. The outputs from $M_A$ do not have such patterns, and it produces more concise captions. Second, the reference captions in Textcaps's tests are relatively much shorter than those from $M_S$ and similar to those from $M_A$. For these reasons, the Textcaps score is low for $M_S$.

It's important to note that without Textcaps, the average result of $M_A$ is 64.06 and that of $M_S$ is 63.63, still suggesting a slightly better or at least competitive performance of AITQE compared to ShareCaptioner. This shows the effectiveness of AITQE as an image-text data-quality enhancer.

# B AVERAGE TOKENS IN GENERATION PHASE

We count the tokens in generation phase and compare AITQE with MLM-Filter and Sharecaptioner.

| Method | Text Tokens | Visual Tokens | Total Tokens | Total inputs for 4 Scores |
|---|---|---|---|---|
| MLM-Filter | 268 | 576 | 844 | 3376 |
| AITQE | 27.15 | 729 | 756.15 | - |

Table 7: Comparison of input tokens between MLM-Filter and AITQE

For MLM-Filter, to get scores of the four criteria, users have to run four generations with different prompts describing the criteria in detail, each time with the same image input. Unlike this, AITQE uses only one relatively shorter prompt to get an overall score. AITQE's intermediate model without caption-rewrite ability (trained with Base + Contrastive Sample) has competitive performance (59.81) compared to MLM-Filter on average (59.69) under the 256K filtering setting, and it can also act as a scorer with early stopping as MLM-Filter can do. Therefore, we compare the input tokens with MLM-Filter in Tab. 7.

The average number of tokens for 4 criteria prompts is 268 (1072 in total), and the visual tokens from clip-vit-large-patch14-336 are 576. Then, the average input tokens to get one score is 844 ($= 268 + 576$), and the total number of input tokens for four scores is 3376 ($= 1072 + 576 \times 4$). In AITQE, the input instructions have only an average of 27.15 tokens, with 729 visual tokens from siglip-so400m-patch14-384, thus a total number of 756.15 tokens as inputs for an overall score, which is 4 times less than 3376.

| Method | Text Tokens (Input) | Visual Tokens (Input) | Total Input Tokens | Average Output Tokens |
|---|---|---|---|---|
| ShareCaptioner | 13 | 576 | 589 | 212.734 |
| AITQE | 27.15 | 729 | 756.15 | 18.326 |

Table 8: Comparison of input and output tokens between ShareCaptioner and AITQE

As shown in Tab. 8, ShareCaptioner takes an input image (576 tokens from clip-vit-large-patch14-336) and a short instruction text "Analyze the image in a comprehensive and detailed manner" (13 tokens). The total input tokens are 589, which is less than AITQE. However, when we average the output tokens of ShareCaptioner in our experiments, the number is 212.734. Meanwhile, AITQE has average outputs of 18.326 tokens with early stopping to get caption rewrite and overall score, which is a significantly smaller number and makes the generation phase of AITQE faster.

## C  SCORE DISTRIBUTIONS OF MLM-FILTER AND AITQE ON 2M DATA

Here we provide the score distributions of MLM-Filter on 2M data and that of AITQE.

In Fig. 7, we can find that although MLM-Filter has a score range of $[1, 100]$, the effective scores are sparse and far fewer than this range suggests. For example, the image-text-matching score has only around 5 valid scores with significant frequency. This number for caption-text-quality is around 8, for object-detail-fulfillment is 11, and for semantic-understanding is 9, which is not ideal for a scorer.

In contrast, Fig. 8 shows the score distribution of AITQE, which has scores ranging from $[1, 10]$. The distribution is similar to that of GPT scores, suggesting good scoring outputs. This figure also explains why using 256K-filtered data can even achieve better results in Tab. 3: the volume of the best quality data in this 2M data pool with scores equal to and greater than 8 is around 300K.

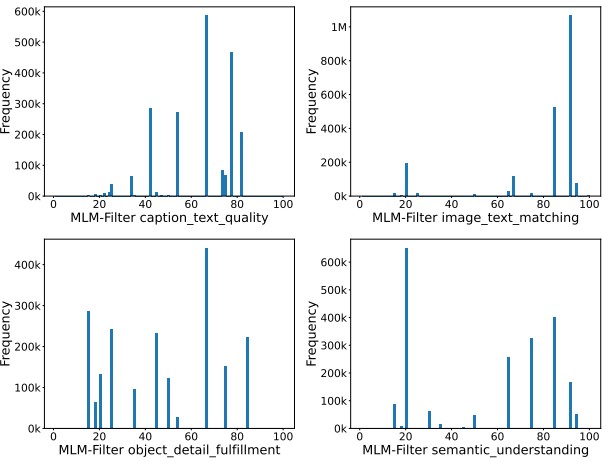

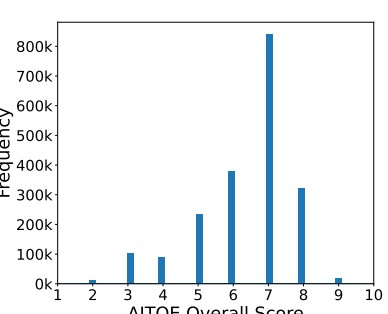

Figure 7: MLM-Filter scores of 2M data

Figure 8: AITQE score of 2M data

## D  EXPERIMENT DETAILS OF SPECS AND HYPERPARAMETERS

We conducted experiments using a cluster of 4 nodes, each comprising 8 NVIDIA H800 GPUs. The training of MLLMs takes different amounts of time due to different data sizes: for 256K with Cauldron (around 890k, same in other MLLMs' training), it takes about 3.5h on 2 nodes; for 558K, about 4.5h; for 2M, about 10h; for 12M, about 42h. The SFT stage of AITQE takes about 19.5h on 4 nodes. Evaluation takes about 2h on 1 node. The hyperparameters are in Tab. 9 and Tab. 10.

Table 9: MLLM training

| Hyperparameter | setting |
| --- | --- |
| global_batch_size | 256 |
| sequence_length | 4096 |
| gradient_clipping | 1.0 |
| LLM_learning_rate | 2e-5 |
| vision_lr | 2e-6 |
| projection_lr | 1e-4 |
| weight_decay | 0 |
| lr_schedule | cosine |
| lr_warmup_ratio | 0.03 |
| deepspeed | ZeRO-2 |

Table 10: AITQE SFT

| Hyperparameter | setting |
| --- | --- |
| global_batch_size | 256 |
| sequence_length | 8192 |
| gradient_clipping | 1.0 |
| LLM_learning_rate | 1e-5 |
| vision_lr | 2e-6 |
| projection_lr | 1e-5 |
| weight_decay | 0 |
| lr_schedule | cosine |
| lr_warmup_ratio | 0.03 |
| deepspeed | ZeRO-2 |

# E   PROMPTS AND SFT DATA CONSTRUCTION

We provide the prompts used to instruct GPT-4o (gpt-4o-2024-08-06), with a JSON schema as the "response_format" parameter for structured output. After data collection, we re-organize the response and create "User-Assistant" conversations as SFT data for AITQE training. For the User instructions in the format "[Image]\n⟨Score Instruction⟩: [Caption]\n⟨Rewrite Instruction⟩\n⟨Format Instruction⟩", we write multiple prompts and randomly choose from them. These are presented in the boxes below.

And in Sec. 4.4, the three intermediate models apart from AITQE are trained with the following data, and these symbols are the same as those in Sec. 3.1:

For Base scorer:

$$D = \left\{ \begin{array}{l} \{(I_1, C_1) : S(I_1, C_1)\}, \\ \{(I_2, C_2) : S(I_2, C_2)\} \end{array} \right\}.$$

For Base scorer + Contrastive Sample:

$$D = \left\{ \begin{array}{ll} \{(I_1, C_1) : S(I_1, C_1)\}, & \{(I_1, C'_{\text{low}}) : S'_{\text{low}}\}, \\ \{(I_2, C_2) : S(I_2, C_2)\}, & \{(I_2, C'_{\text{high}}) : S'_{\text{high}}\} \end{array} \right\}.$$

For Base scorer + Rewrite Caption:

$$D = \left\{ \begin{array}{l} \{(I_1, C_1) : S(I_1, C_1)\}, \\ \{(I_2, C_2) : (C'_{\text{high}}, S(I_2, C_2))\} \end{array} \right\}.$$

---

**Scoring Prompt and JSON schema**

**System Role:**
Your role is to serve as an impartial and objective evaluator of image captions. Please evaluate the given caption with the following criteria and provide each criteria score and explanation, then an overall score and explanation.

**Evaluate the caption based on:**

1. **Text Quality:** Grammar, vocabulary, fluency, readability, length, structure

2. **Image-Text Matching:** Accuracy in representing key elements and overall theme

3. **Object Detail:** Detailed descriptions of objects (color, size, position, shape, material, etc.)

4. **Semantic Understanding:** Additional information beyond visual content

5. **Text/Chart Description:** If the image contains significant text or charts, evaluate how well the caption describes this content

**Output Format:**

- [criteria] Score: ⟨integer from 1 to 10⟩
- [criteria] Explanation: ⟨explanation of your evaluation⟩

- - - - - - - - - - - - - - - - - - - - - - - - - - - - - - - - - - - - - - -

**JSON Schema for Structured Outpus with the following properties:**

- Overall Score (integer) and Overall Explanation (string)
- Text Quality Score and Explanation
- Image-Text Matching Score and Explanation
- Object Detail Score and Explanation
- Semantic Understanding Score and Explanation
- Text/Chart Description Score and Explanation
- *(All fields are required, and no additional properties are allowed.)*

---

**Rewrite Caption Prompt and JSON schema**

**System Role:**
Your role is to serve as an assistant that generates captions of good quality for images. You will be given an original caption of low quality, and you need to generate a one-sentence high-quality re-caption. Output:

1. The generated high-quality re-caption
2. Give a high score of your re-caption based on the provided criteria

**Evaluation Criteria:** Same as *Scoring Prompt*
**Output Format:**

- Recaption: ⟨your one-sentence high-quality re-caption⟩
- [criteria] Score: ⟨score of your re-caption, integer from 1 to 10⟩
- [criteria] Explanation: ⟨explanation of your evaluation⟩

-----

**JSON Schema for Structured Outpus with the following properties:**

- Recaption (string)
- Others Same as *Scoring Prompt*

---

**Generate Bad Caption Prompt and JSON schema**

**System Role:**
Your role is to serve as an assistant that generates bad captions of given images. You will be given an original caption of relatively good quality, and you need to generate an one-sentence bad-quality caption. Output:

1. The generated bad-quality caption, could be inaccurate, image-text mismatching, or irrelevant information
2. Give a low score of your bad-quality caption based on the provided criteria. Give explanation; use varying words rather than repeating the instructions.

**Evaluation Criteria:** Same as *Scoring Prompt*
**Output Format:**

- Badcaption: ⟨your one-sentence bad-quality caption⟩
- [criteria] Score: ⟨core of your bad-quality caption, integer from 1 to 10⟩
- [criteria] Explanation: ⟨explanation of your evaluation⟩

-----

**JSON Schema for Structured Outpus with the following properties:**

- Badcaption (string)
- Others Same as *Scoring Prompt*

**Re-organized SFT data**

**User:** [Image]
Evaluate the caption: [Caption].
If score is low, give a short summary of the image.
Respond using a dictionary structure.

**Assistant:**
"{"Recaption": "...",
"Overall Score": "...⟨Overall⟩", "Overall Explanation": "...",
"Text Quality Score": ..., "Text Quality Explanation": "...",
"Image-Text Matching Score": ..., "Image-Text Matching Explanation": "...",
"Object Detail Score": ..., "Object Detail Explanation": "...",
"Semantic Understanding Score": ..., "Semantic Understanding Explanation": "...",
"Text/Chart Description Score": ..., "Text/Chart Description Explanation": "..."}"

**User Instruction Sets**

**Score Instruction:**
Evaluate the caption; Rate the effectiveness of the caption; Assess the quality of the caption; Analyze the caption's impact; Grade the caption; Measure the caption's effectiveness; Appraise the caption; Judge the caption's quality; Determine the caption's score; Quantify the caption's effectiveness; Rank the caption; Gauge the caption's strength; Review and score the caption; Provide a rating for the caption; Critique and score the caption; Assign a value to the caption; Weigh the merits of the caption; Calculate the caption's score; Estimate the caption's effectiveness; Examine and rate the caption

- - - - - - - - - - - - - - - - - - - - - - - - - - - - - - - - - - - - - - - -

**Rewrite Instruction:**
If score is low, give a recap of the image; If low score, recaption the image; if score is low, give a short summary of the image; if score is low, give a short description of the image; For low-scoring images, provide a concise overview; When the score is low, briefly describe the main elements of the image; If the score is below threshold, generate a new caption; For poorly scored images, summarize the key visual components; If the image scores low, provide a succinct description of its content; When faced with a low-scoring image, offer a brief caption of what it depicts.

- - - - - - - - - - - - - - - - - - - - - - - - - - - - - - - - - - - - - - - -

**Format Instruction:**
Respond using a dictionary structure; Format your answer as a dict; Present the result in a key-value pair format; Output the response as a dictionary; Provide the answer in a dict-like structure; Use a dictionary format for your reply; Structure your response as a dict; Return the information in a key-value format; Organize the answer in a dict format; Express the result using dictionary notation; Format the output as a key-value dictionary; Give the answer in a dict structure; Represent the response using a dictionary; Reply with a dict-formatted answer; Construct your response as a dictionary; Arrange the information in a dict format; Present the data in a key-value dictionary; Formulate your answer as a dict; Deliver the response in dictionary format; Compose your reply using a dict structur.

