# OpenReview forum: "Beyond Filtering: Adaptive Image-Text Quality Enhancement for MLLM Pretraining"
_ICLR.cc/2025/Conference — ICLR 2025 Conference Withdrawn Submission_

### Official Review · Reviewer_P8fx · 2024-11-02

**Soundness:** 2
**Presentation:** 2
**Contribution:** 2
**Rating:** 3
**Confidence:** 4

**Summary:**

This paper proposes an adaptive approach to improve data quality for training MLLMs. The method incorporates two main techniques: text rewriting and contrastive sample learning. GPT4o is used to recaption data and identify low-quality samples. The text rewriting strategy aims to preserve data volume without excessive sample removal.

**Strengths:**

- The paper addresses a critical challenge in MLLM training data quality enhancement, which is essential for improving model performance and scalability.
- Results demonstrate the importance of dataset size and quality, highlighting the trade-offs involved in data refinement.

**Weaknesses:**

### 1. Limited Novelty and Comparison with MLM-filter
- **Unclear Contribution Over MLM-filter**: The proposed method borrows several techniques from MLM-filter (Ln 158, Ln 193). However, the paper lacks a detailed comparison with MLM-filter both theoretically and experimentally, which is essential to establish novelty.
- **Ablation Study Insufficiencies**: In Table 6, ablation results suggest that individual techniques, i.e., contrastive sampling and caption rewriting, do not improve performance independently, as they downgrade results when applied alone. This raises concerns about the standalone effectiveness of these techniques.
- **Marginal Improvement Over MLM-filter**: As seen in Table 3, the performance difference is minimal (0.05) when using 558k samples, which challenges the claim that AITQE can be "scaled efficiently." This discrepancy suggests limited gains over MLM-filter and raises doubts about scalability claims.
- **Contradictory Analysis**: The discussion in Ln 335-349 appears to conflict with the motivation of AITQE. A primary claim is the enhancement of low-quality data, yet the analysis suggests that data removal (e.g., using 558k samples out of 2M) is inevitable, contrary to the stated goals of AITQE.

### 2. Insufficient Comparison and Incomplete Experiment Design
- **Missing Comparison with MLM-filter in Key Experiments**: Experiments in Table 4 do not include MLM-filter for comparison. Given that Table 3 indicates only a small improvement by AITQE, a broader comparison in other scaling scenarios (like in Table 4) is essential to demonstrate clear advantages.
- **Section 3.3's Contribution is Minimal**: Tables 1 and 2 are not directly related to the primary contributions of the paper and could be moved to the appendix to improve focus on the main findings. Taking a step back, the experimental design in Section 3.3 is also quite incomplete. Many experimental details could affect the results, for example, dataset size, random seed, learning rates and learning scheduler etc. More importantly, only two runs is not enough to conclude that two stage training is worse. Lastly, the difference between two run in Tab. 2 is also relatively large.

### 3. Experiments Do Not Support Proposed Method’s Advantage
- **Weak Comparison with ShareCaptioner**: The comparison in Sec. 4.3, conducted with ShareCaptioner, lacks context and significance, as both ShareCaptioner and AITQE employ GPT for recaptioning. The comparison appears to show only that GPT4o has improved over earlier GPT4V, rather than highlighting advantages of AITQE specifically.
- **Unexplained Ablation Results in Section 4.4**: Ablation studies lack a clear explanation for why individual techniques underperform in isolation and why their combination yields improvements. Presenting numerical results alone does not provide sufficient insight into the underlying mechanisms.

### 4. Other Observations
- **Limitation Description**: Regarding the absence of CoT as a limitation seems strange.

**Questions:**

please see weakness points above.

---

### Official Review · Reviewer_erp5 · 2024-11-02

**Soundness:** 3
**Presentation:** 3
**Contribution:** 3
**Rating:** 6
**Confidence:** 2

**Summary:**

This paper introduces a highly-perform image-text pair quality enhancement and quality scoring method：the Adaptive Image-Text Quality Enhancer (AITQE). This method surpasses existing approaches on benchmark datasets using the same raw data.

**Strengths:**

1.The paper is relatively well-written, the data-construction pipeline is illustrated clearly.

2.The experiments and analysis are relatively sufficient with multiple benchmarks and various experimental settings.

3.The idea of the contrastive sample learning strategy seems to be novel，maintaining relatively large amount of high-quality images while enhancing the model’s ability on alignment quality scoring and caption enhancing.

**Weaknesses:**

1.One problem remains to me is that the verification of the original 2-stages training strategy is “unstable” (the 5th page).  Since vast amount of LMMs is trained through this way, conducting only the two-times experiments and using the difference in their results as evidence for the stability of the training framework may not be sufficiently convincing. A more robust analysis should be given to prove your conclusion.  Besides, why is your training strategy “stable” enough，there should be supplementary experiments on this.

2.Some more compare methods should be involved into the Compare AITQE with captioner experiments (the 8th page), since caption enhancement is one of the largest contributions announced in the paper.

**Questions:**

Please see the weakness.

---

### Official Review · Reviewer_CTSD · 2024-11-03

**Soundness:** 3
**Presentation:** 3
**Contribution:** 3
**Rating:** 5
**Confidence:** 3

**Summary:**

This paper introduces an adaptive image-text quality enhancer (AITQE) which designed to assess and enhance the quality of input image-text pairs.
First, existing image-text pair data are assessed by multimodal large language models (MLLMs) and their contrastive samples are collected.
The proposed AITQE consists of SigLIP and Qwen-2-7B is trained using the collected data, then the model can assess and enhance low quality data.
In experiments, the model trained with the AITQE surpasses random sampling and existing approaches on benchmarks datasets using the same raw data.

**Strengths:**

- AITQE effectively enhances low quality image captions automatically.
- AITQE's filtering ability is explained in section 4.1.
- Section 4.2 shows that AITQE scales effectively with increasing volumes.

**Weaknesses:**

- I'm not sure comparisons in table 1 and table 2 are fair or not. They are evaluated on different benchmarks (total 5 and 8 respectively) so it is unfair to directly compare the values, because using more benchmarks will give a small discrepancy among different runs in terms of average.

**Questions:**

- In table 4, is there any other methods rather than random sampling?
- Why are the results of 256K-AITQE in table 4 and AITQE in table 6 differ?
- How the scoring prompt and contrastive sampling prompt in figure 2 have been selected. Could the authors provide any idea or reasons behind?
- The data collection is conducted by GPT-4o and AITQE consists of SigLIP and Qwen2-7B which may have lower performance than GPT-4o. In terms of distillation, AITQE, the student, is assisted by a large and better teacher model. So what if data collection is conducted by older/smaller models like Llama2 or the same SigLIP + Qwen2-7B model? Will it still work?

---

### Official Review · Reviewer_CWNN · 2024-11-03

**Soundness:** 4
**Presentation:** 4
**Contribution:** 3
**Rating:** 5
**Confidence:** 5

**Summary:**

This paper presents the Adaptive Image-Text Quality Enhancer (AITQE) for multimodal large language model (MLLM) pretraining. We know that data quality plays an important role for MLLM. To this end, the authors generate many positive and negative data to develop AITQE. Then, AITQE is utilized to continuously rewrite and filter high-quality data for pretraining. Experimental results show its superiority over existing methods in leveraging data and scaling with data volume.

**Strengths:**

The strengths of the submission are as follows:

S1. The studied task, MLLM, is very important and useful.

S2. The idea of generating high-quality data is also technical sound.

S3. This paper is also well-written and is easy to follow.

**Weaknesses:**

There are several major weaknesses.

1. From the motivation side. It is important for us to generate high-quality data for model training. However, this idea have been discussed by some better caption techniques, like ShareGPT4V, ALLaVA, and LLaVA-OV. I think the authors should better demonstrate their advantages compared with these related works.

2. From the efficiency side. As reported in the methodology, the authors utilize GPT-4o to generate many data, which means they should call API many times.

3. From the experimental side. Although the author has conducted numerous experiments for verification, we know that MLLM focuses on the final performance. I want to know if the effectiveness of the authors' solution will be diluted by various other techniques, such as better captioning and dynamic high-resolution technology.

**Questions:**

Following are my core concerns and questions.

C1. Is there any connection between your method and Better captioning technology? Can your method replace the mainstream 1.5-stage pre-training technology (LLaVA-OV, ALLaVA), or can your solution work together with these works to achieve better performance? I want to know if the performance of AITQE will be diluted by other methods.

C2. The authors have conducted some experiments. However, some reasoning-based datasets are missed, for example, MM-Vet. I would like to know whether the reasoning performance is boosted.

C3. The authors use a lot of prompts when using GPT-4o. I want to know what are the advantages of this design compared to a fixed prompt.

C4. We know that GPT-4o is a closed-source model. The use of GPT-4o is not always easy (network & token per second limitation). Can we replicate your ideas through some open-source models, such as QWEN2-VL?

C5. In the author's experiment, we can see that more data is not necessarily better. Can the author provide some more suggestions on scaling law? For example, for a total of $x$ data, selecting $y$% of the data for training yields the best performance. Will this conclusion change with different $x$?

Justification for rating:

Although this paper is useful, similar ideas have also been discussed in related papers. Besides, some important conclusions are also missed. Therefore, I decide to offer an negative score before rebuttal.

---

### Official Review · Reviewer_BfdY · 2024-11-03

**Soundness:** 2
**Presentation:** 1
**Contribution:** 1
**Rating:** 3
**Confidence:** 4

**Summary:**

The paper proposes to enhance texts in image-language pair data to improve image-language tasks. Instead of removing (filtering) tentatively-low-quality text descriptions of the corresponding images, the paper proposes to enhance low-quality text description to increase 'utility' of the data. In details, the proposed method first generates contrastive image-text pairs. The method measures each image-text pair quality with GPT-4o ranging 1-10. If the quality measured by GPT-4o is high (low), the method uses GPT-4o again to generate low (high) quality text description for the corresponding image. The model combined SigLIP vision encoder and Qwen2 language model is trained with the contrastive samples with their scores. The proposed method shows better performance than filtering and random sampling method in Document-related dataset.

**Strengths:**

The major novel contribution of the paper is to generate contrastive samples for model training. The model is trained with both high- and low-quality captions (with the scores) to comparably learn good and bad caption.

**Weaknesses:**

Proposed data enhancement is highly dependent on the GPT-4o which degenerated novelty. And, the performance is not considerable better than filtering algorithms or random sampling. In Table 3, the proposed method is mainly performing better than filtering algorithms in document (OCR) related dataset. In the other dataset, the proposed method does not perform better better than the other methods. The reviewer asserts that this is mainly because of GPT-4o’s OCR performance, not the contrastive samples which the paper’s main contribution. The same tendencies are shown in Table 4.

To show the effectiveness of the main contribution (contrastive samples), the author needs to compare with data enhancing method using the same GPT-4o without contrastive samples.

Clearer explanation is required and some analysis of the experiments are misleading. See below,
- Line 281, “The foundation …”, This statement is ambiguous and no ground for this statement. What is enhancer? data enhancer? What is targeted improvement? Discerning data quality is `the` foundation of enhancer? As far as the reviewer understands, this statement is incorrect.

- Most of the experimental setups are unclear. What is 12M in L 288? “The datasets of 256K, 558K and 2M samples …” contradicts L 284...

- No explanation of inference. what is the input and output of the model in inference time.

- In Table 3, why compared with LLaVA-558k? It is contradicting with statements In L525, pre-experiments.

- In 3.3 Assess the AITQE model, the paper did not show SigLIP + Qwen2 with Cauldron evaluation setup is 'more' stable than LLAVA with LLAVA 558K stage-1 and 664K stage-2 evaluation setup.

- Please add reference of ShareCaptioner

- Recommended remove statement L366. The data point is too small and the performance difference between proposed and random sampling is almost the same as increasing size of samples.

- There are more ..

**Questions:**

Does ShareCaptioner use GPT-4o?

---

### Official Review · Reviewer_7e73 · 2024-11-04

**Soundness:** 2
**Presentation:** 2
**Contribution:** 2
**Rating:** 5
**Confidence:** 5

**Summary:**

This paper proposes a data filtering strategy. Unlike the traditional method of discarding low-quality data and retaining high-quality data, it can enhance low-quality data to ensure high quality and diversity of data. Experimental results show that using this grade system to filter training data can have a certain impact on the results.

**Strengths:**

1. The AITQE method proposed in this paper can automatically filter or rewrite image-text pairs based on their assessed quality. It has certain practical value in pre-training scenarios.

2. The writing of this article is quite fluent, and with appropriate illustrations. Figures 1 and 2 both intuitively express the innovation of the article. It is very easy for readers to understand.

3. The experimental settings are listed in details, which has strong reproducibility.

**Weaknesses:**

1. The evaluation method of the experiment is not rigorous enough. First, the quality improvement of Figure 4 is an inevitable result and cannot be regarded as a contribution. For example, find a golden text result, and then add the text of each Image-Text pair to it, and the quality score will definitely be higher than that of raw data. It is important to ensure diversity while improving quality, as the author said in the introduction. I haven't seen the author's analysis on diversity.

2. The training results are not very satisfactory. Table 3 shows that AITQE is indeed better than MLLM filtering in terms of overall performance, but this improvement is all from Textcaps. On other datasets, the two are just comparable or even inferior. Leading by a large margin on only one dataset, thus leading in average score, will cause generalization problems.

**Questions:**

If the author cannot improve the performance on each dataset, I am not sure if the author can narrow the topic of the article to data filtering for text captions instead of general? I admit that this article has made a great contribution, but it is mainly on Textcaps. This modification may better clarify the advantages and applicable scenarios of AITQE. Also, the authors can check the Weaknesses, and address them point-by-point in the response, which would be helpful.

---

### Official Review · Reviewer_Ybdh · 2024-11-08

**Soundness:** 3
**Presentation:** 4
**Contribution:** 3
**Rating:** 6
**Confidence:** 3

**Summary:**

This paper tackles the challenge of improving data quality in multimodal large language models (MLLMs) by proposing the Adaptive Image-Text Quality Enhancer (AITQE). Unlike traditional methods that discard valuable data due to poor image-text alignment, AITQE adaptively refines low-quality pairs by minimally rewriting text and using a negative sample learning strategy to enhance evaluation capabilities. Experimental results show that AITQE outperforms existing approaches, achieving better data efficiency and scalability, making it a promising solution for optimizing image-text pair quality in MLLM training.

**Strengths:**

The paper highlights several strengths of the proposed Adaptive Image-Text Quality Enhancer (AITQE). It introduces an innovative solution to improve data quality in multimodal large language model (MLLM) training by dynamically assessing and enhancing image-text pairs, offering a significant improvement over traditional methods that discard valuable data. AITQE’s text rewriting mechanism adjusts low-quality pairs with minimal data loss, preserving the original text distribution and maximizing resource utilization. Additionally, the integration of a negative sample learning strategy enhances model robustness, helping it better differentiate between high- and low-quality pairs. The model demonstrates strong scalability with large datasets and efficient data utilization. Experimental results across various benchmarks validate AITQE's effectiveness, positioning it as a promising solution for optimizing multimodal data quality in MLLMs.

**Weaknesses:**

The paper has areas for improvement that could enhance its impact.

 1) It lacks a thorough analysis of potential trade-offs in text rewriting, such as shifts in semantics or bias, which could affect downstream tasks.

2) The criteria for selecting low-quality samples in the negative sample learning strategy are also under-explained, raising questions about AITQE’s generalizability.

3) Scalability is claimed, testing on extremely large datasets could further substantiate these claims. Additionally, the comparison to a broader range of enhancement methods and testing across varied domains would better establish its versatility.

4) A discussion on potential computational overhead could clarify the impact of AITQE’s components on training efficiency. Addressing these aspects would improve AITQE’s robustness and applicability in multimodal data enhancement.

**Questions:**

Work on the weaknesses

---

### Note · Authors · 2024-11-13

I have read and agree with the venue's withdrawal policy on behalf of myself and my co-authors.